# Single-cell transcriptomic analyses of tumor microenvironment and molecular reprograming landscape of metastatic laryngeal squamous cell carcinoma

Yuanyuan Sun [1], Sheng Chen[2], Yongping Lu[3], Zhenming Xu [4✉], Weineng Fu [1✉] & Wei Yan [5,6✉]

Laryngeal squamous cell carcinoma (LSCC) is a malignant tumor with a high probability of metastasis. The tumor microenvironment (TME) plays a critical role in cancer metastasis. To gain insights into the TME of LSCC, we conducted single-cell RNA-seq (scRNA-seq) on samples collected from LSCC patients with or without lymphatic metastasis. The stem and immune cell signatures in LSCC suggest their roles in tumor invasion and metastasis. Infiltration of a large number of regulatory T cells, dysplastic plasma cells, and macrophages that are at the early development stage in the cancerous tissue indicates an immunosuppressive state. Abundant neutrophils detected at the cancer margins reflect the inflammatory microenvironment. In addition to dynamic ligand-receptor interactions between the stromal and myeloid cells, the enhanced autophagy in endothelial cells and fibroblasts implies a role in nutrient supply. Taken together, the comprehensive atlas of LSCC obtained allowed us to identify a complex yet unique TME of LSCC, which may help identify potential diagnostic biomarkers and therapeutic targets for LSCC.

[1] Department of Medical Genetics, China Medical University, Shenyang 110122, China. [2] Department of Laboratory Animal Science, China Medical University, Shenyang 110122, China. [3] NHC Key Laboratory of Reproductive Health and Medical Genetics, Shenyang 110122, China. [4] Department of Otolaryngology, the Fourth People's Hospital of Shenyang City, Shenyang 110031, China. [5] The Lundquist Institute for Biomedical Innovation at Harbor-UCLA Medical Center, Torrance, CA 90502, USA. [6] Department of Medicine, David Geffen School of Medicine at UCLA, Los Angeles, CA 90095, USA. ✉email: zhenmingxu@cmu.edu.cn; wnfu@cmu.edu.cn; wei.yan@lundquist.org

L aryngeal squamous cell carcinoma (LSCC) is one of the common malignant tumors of the head and neck. The early symptoms of LSCC are not obvious and consequently, more than half of the patients, upon diagnosis, are already at the mid or late stage of tumor progression with infiltration and lymphatic metastasis[1]. Recurrence and metastasis are two important factors affecting the five-year survival rate in LSCC patients[2,3]. Therefore, early diagnosis and prevention of metastasis are key to the treatment of LSCC. Since the concept of the tumor microenvironment (TME) emerged, it has been widely acknowledged that TME plays a critical role in tumor progression. The TME refers to a complex, heterogeneous composition of infiltrating immune and resident host cells, secreted factors, and extracellular matrix. Dynamic and reciprocal interactions between cancer cells and components of the TME lead to conditions that favor cancer cell survival, local invasion, and metastatic dissemination. For example, the TME can promote angiogenesis to overcome the hypoxic and acidic microenvironment; diverse adaptive and innate immune cells infiltrate the tumor to exert either pro- or anti-tumorigenic functions. Therefore, it is of great significance to study tumor-specific TME as it helps us not only understand tumor progression but also identify novel targets for diagnostics and therapeutics. Indeed, TME-based precision medicine has drawn great attention over the last decade. In particular, personalized immunotherapies for lung cancer, ovarian cancer, and pancreatic cancer have widely been used in clinical practice, and consequently, the quality of life and five-year survival rate of cancer patients have improved greatly[4–6]. However, highly variable outcomes among patients with various tumor types suggest an incomplete understanding of the TME. The TME of LSCC with lymphatic metastasis remains unclear at present, and so is the role of TME in tumorigenesis and metastasis of LSCC.

To elucidate the TME of LSCC, we carried out the present study with the following goals: (1) Obtain a single-cell atlas of LSCC with lymphatic metastasis using scRNA-seq to determine the cellular heterogeneity in LSCC. (2) Identify specific cell subclusters associated with LSCC metastasis and determine the gene signatures that may serve as potential biomarkers for early diagnosis and treatment. (3) Analyze copy number variations (CNVs), transcription factors (TFs), and signaling pathways regulating each subcluster of the epithelial cells to explore the mechanisms of tumorigenesis. (4) Analyze the heterogeneity, function, developmental trajectories, and key TFs of immune cells to reveal the mechanism underlying the immune escape of LSCC. (5) Define the cell-cell interaction networks to gain insights into the complex cell-cell communications in the LSCC TME. To accomplish these goals, we performed scRNA-seq analyses on samples from LSCC patients with lymphatic metastasis using the 10X Genomics single-cell platform followed by in-depth bioinformatic analyses. The comprehensive atlas of LSCC obtained allowed us to identify a complex yet unique tumor microenvironment of laryngeal squamous cell carcinoma (LSCC) with lymphatic metastasis.

## Results

### Single-cell atlas and cellular heterogeneity in metastatic LSCC.
Four types of samples were collected from six LSCC patients undergoing surgery (Supplementary Table S1), including tumor in situ (T), normal laryngeal mucosal epithelia adjacent to the tumor (N), margins of cancer (R), and lymph nodes with metastasized cancer cells (L). Histology of all of the samples was examined using HE-staining of paraffin sections (Supplementary Fig. S1). Single-cell suspensions were prepared immediately after sample collection during surgery followed by scRNA-seq and in-depth bioinformatic analyses (Fig. 1a). Quality control of the

scRNA-seq data was performed by analyzing unique molecular identifier (UMI) numbers, gene numbers, and the percentage of mitochondrial genes per cell (Supplementary Fig. S2a–c), and cell-cycle-related genes appeared to have no effects on cell cluster analyses (Supplementary Fig. S2d–e). A total of 89,406 single cells from all of the ten samples, including T, R, N, and L, were captured and sequenced (Supplementary Table S2). A total of 25 subclusters were identified and 7 cell types were annotated, including epithelial-derived cells (EpCs), myeloid cells, T cells, B cells, NK cells, endothelial cells, and cancer-associated fibroblasts (CAFs) (Fig. 1b). The cell identity was further verified by t-SNE plots (Fig. 1c) using known marker genes, including *EPCAM* for epithelial-derived cells, *CD3* for T cells, *CD19* for B cells, *CD33* for myeloid cells, *CD56* for NK cells, *VEGFC* for endothelial cells, and *a-SMA* for fibroblasts, as well as immunofluorescent staining of these marker proteins[7] (Fig. 1d).

To reveal the cellular heterogeneity of LSCC with lymph node metastasis, we re-clustered four major cell types, including EpCs, T cells, B cells, and myeloid cells. We further divided T samples into T1 and T2, representing LSCC with and without lymph node metastasis, respectively. Eleven subclusters in epithelial-derived cells, ten subclusters in T cells, nine subclusters in B cells, and ten subclusters in myeloid cells were identified (Fig. 1e). Significant differences in cell numbers and cell types were observed among all of the five types of samples (T1, T2, R, N, and L), especially in T cells and myeloid cells (Fig. 1e, f). These data reveal a highly complex TME that is constantly changing with the progression of LSCC, thus validating the notion that the "tumor microenvironment is not just a silent bystander, but rather an active promotor of cancer progression"[8].

### Correlations of stem and immune cell features in LSCC with invasion and metastasis.
Numerous copy number variations (CNVs), including deletions (3q, 5p, 5q, 13p, and 13q) and copy number gains (3p, 6q, 7q, 11q, and 12q), were detected in EpCs (Fig. 2a). The CNV patterns allowed for classification of the EpCs into malignant and non-malignant groups (Fig. 2a). Further clustering analyses revealed 11 subclusters in the five types of tissues (L, T1, T2, N, and R) (Fig. 2b). Clusters C0, C5, C6 and C10 mainly existed in normal laryngeal mucosal epithelia (N); clusters C4 and C8 in lymph nodes with metastasized LSCC (L); clusters C1, C2, C7 and C9 in tumor in situ (T); cluster C4 in both T and L; clusters C3, C4, C5, C6 and C10 in tumor margins (R); clusters C1 and C2 in tumor with and without metastasis (T1 and T2). Based on differentially expressed genes (DEGs), the functions of each subcluster were further investigated using Gene Set Enrichment Analysis (GSEA)[9]. The upregulated genes in clusters C3, C4, C5 and C10 were mostly those that promote proliferation (with GO terms of "mitotic sister chromosome segregation, DNA replication, and DNA strand elongation") and energy production (with GO terms of "NADH dehydrogenase, respiratory electron transport chain, etc.") (Fig. 2c, d; Supplementary Table S3), whereas the upregulated genes in clusters C7 (Fig. 2c, d; Supplementary Table S3) contained those usually expressed in early embryos ("embryonic skeletal system morphogenesis and development"). Both L and T samples displayed downregulated genes that control the extracellular matrix, as compared to the N and R samples (Fig. 2c, d; Supplementary Table S3). These results suggest that the malignant cells in both L and T samples have higher proliferative activity, lower differentiation and immune chemotaxis, which may account for their enhanced capability of invasion and metastasis.

Developmental trajectory analyses revealed that subclusters C2, C7 and C8, which displayed stem cell features, were the initial cell types correlated with a higher degree of malignancy and

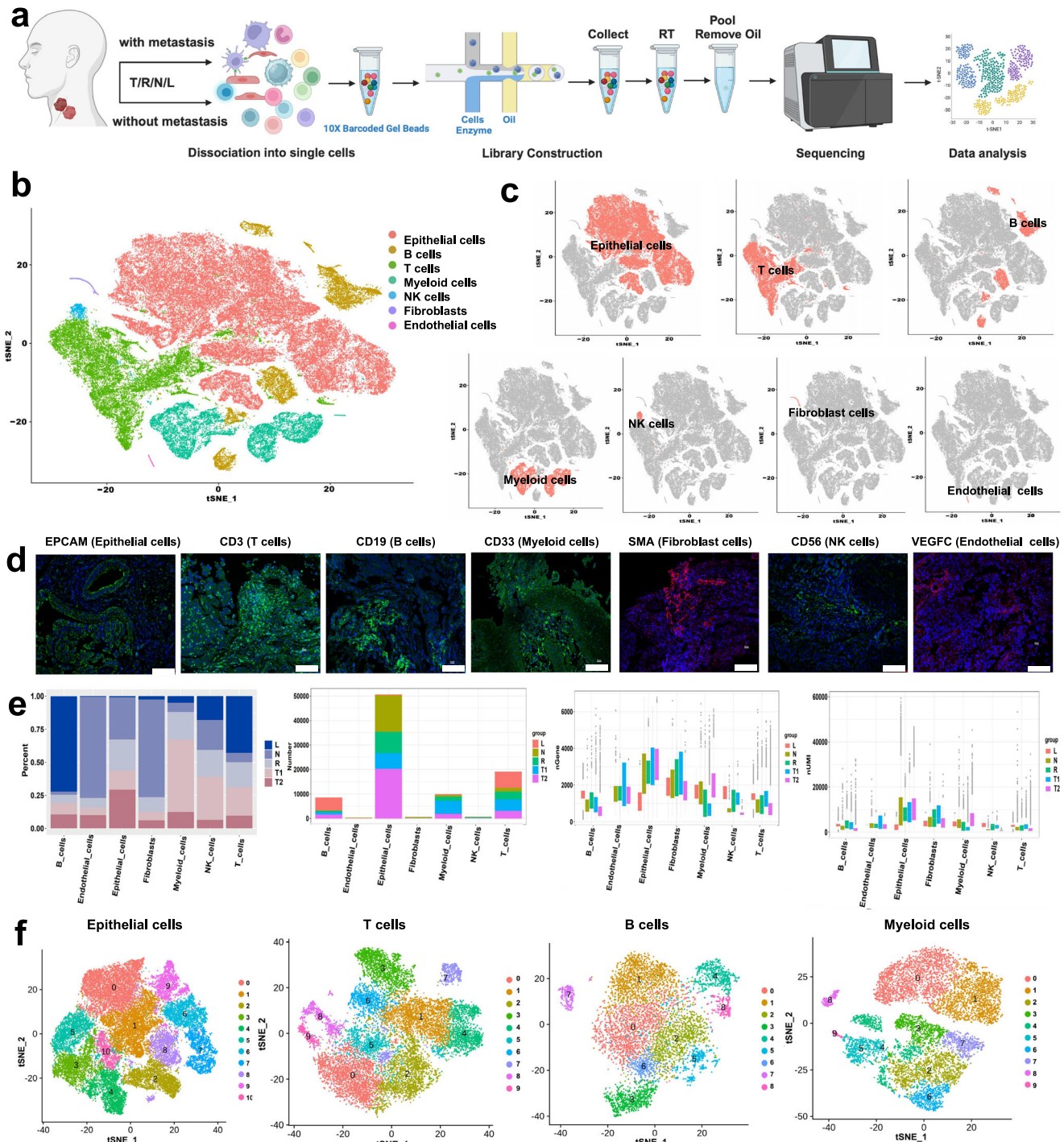

**Fig. 1 A single-cell atlas and transcriptional heterogeneity of LSCC with lymphatic metastasis. a** Diagram showing the workflow of the present study. **b** t-SNE plots showing the seven major cell types identified in four types of tissue samples analyzed in this study. **c** t-SNE plots showing marker gene expression in each of the seven cell types identified in the four types of tissue samples analyzed in this study. Marker genes for epithelial-derived cells: *KRT15, KRT18, KRT19* and *EPCAM*; Marker genes for T cells: *CD2, CD3D, CD3E* and *CD3G*; Markers for B cells: CD19, CD79A, CD79B; Marker genes for myeloid cells: *CD33, CD68, CD1E, LYZ* and *LAMP3*; marker genes for NK cells: *CD56, CD16, NKP46* and *NKP30*; Marker genes for endothelial cells: *VEGFR, TEK* and *CD54*; marker genes for fibroblasts: *alpha-SMA, FAP*, and *S100A4*. **d** Immunofluorescent detection of the seven marker proteins in LSCC tissue cross-sections. Scale bars = 100 μm. **e** Cellular composition, and the numbers of cells, genes, and unique molecular identifiers (UMIs) of all cell types in the different types of samples examined in the present study. **f** t-SNE plots showing the subclusters identified in epithelial/malignant cells, myeloid cells, T cells, and B cells.

metastatic potential and lower keratinization (Fig. 3a). Consistently, SCENIC analyses[10] identified several key transcriptional factors (e.g., *SOX2, TWIST1, HOXC10*, etc.) in subclusters C7 and C8, which are known to be related to stem cell activities (Fig. 3b; Supplementary Fig. S3). To identify the potential biomarkers of LSCC invasion and metastasis, we performed Ingenuity Pathway Analyses (IPA). The most relevant functions identified included cancer, organismal injury or abnormalities, endocrine disorders,

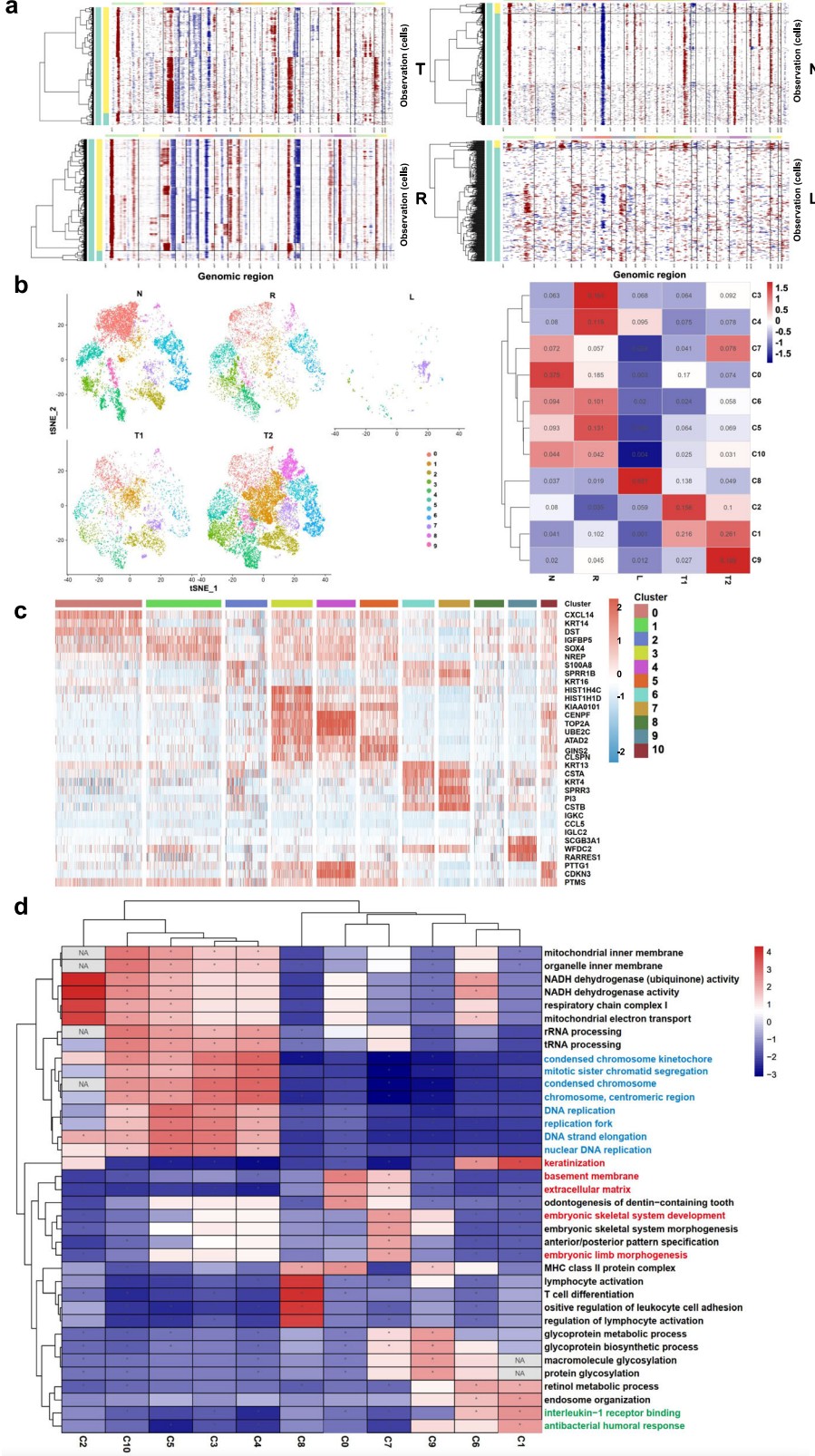

**Fig. 2 Epithelial-derived cell transcriptional heterogeneity of LSCC with lymphatic metastasis. a** Chromosomal landscape of inferred CNVs distinguishing malignant epithelial-derived cells and non-malignant epithelial-derived cells from different types of samples based on the scRNA-seq data. The references are T cells and B cells; chromosomal amplifications are shown in red and deletions in blue. **b** t-SNE plots and heat map showing the distribution of the eleven epithelia-derived cell subclusters in the four tissue types. **c** Heatmap showing the top differentially expressed genes (DEGs) in eleven epithelial-derived cell subclusters. **d** GSEA results showing the activated and suppressed pathways in the eleven epithelial-derived cell subclusters.

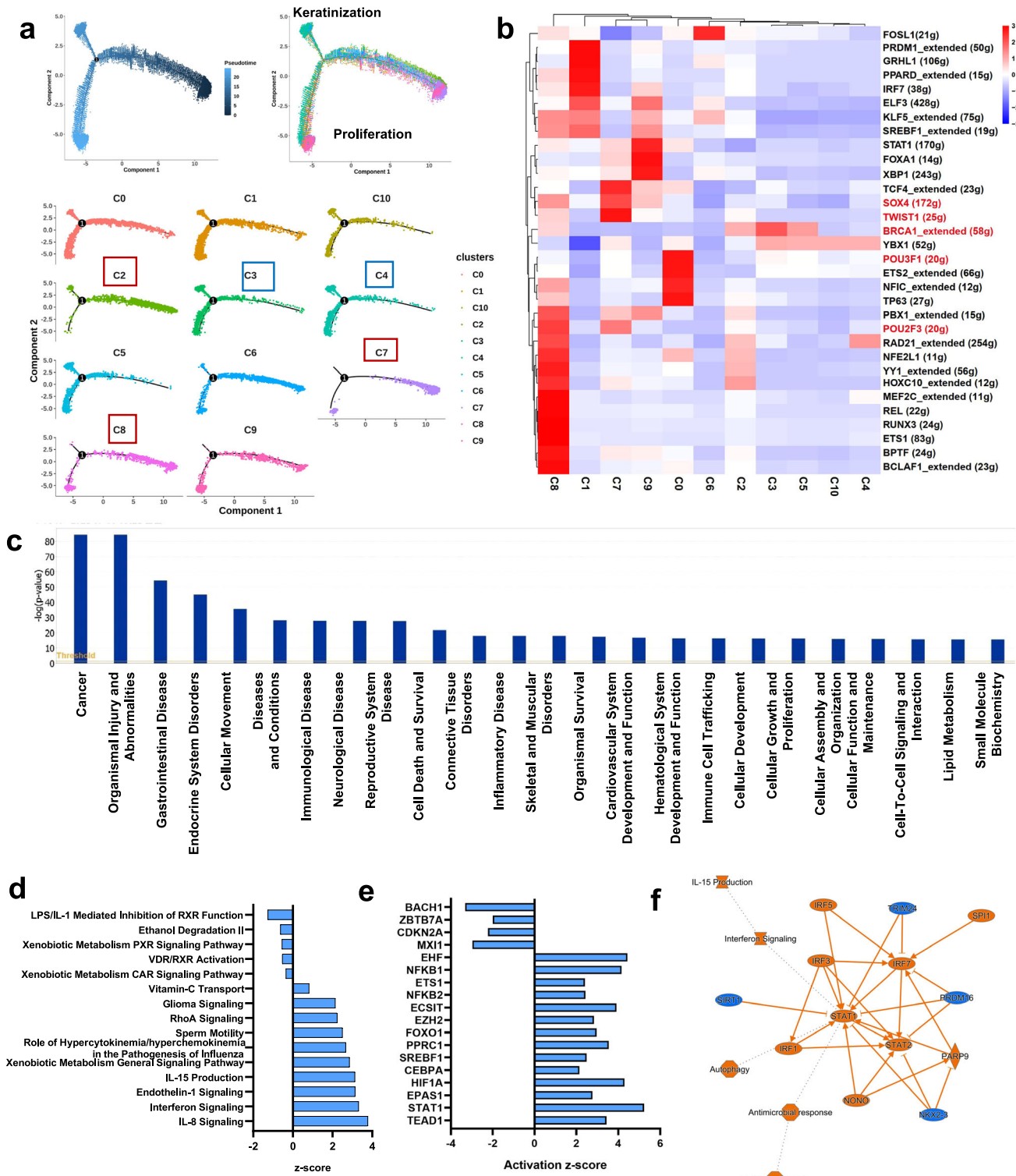

**Fig. 3 The stem and immune cell features in the malignant epithelial cells in LSCC. a** Developmental trajectory of epithelial-derived cells, as revealed by pseudotime analyses (Top two panels). Developmental stages (proliferation, keratinization and migration) of the 11 subclusters in the EpCs are also shown (Lower 11 panels). **b** Heatmap showing the activity of transcriptional factors (TFs) in each of the twelve subclusters of EpCs. **c** Diseases and functions enrichment of DEGs. **d** Pathway enrichment analyses of DEGs between malignant and non-malignant epithelial cells. **e** Predicted activated and inhibited upstream TFs in malignant epithelial cells in LSCC. **f** Graphical summary of ingenuity pathway analyses showing the most affected regulator and effectors in malignant epithelial cells in LSCC.

dermatological conditions, metabolic disease, cell death and survival, infectious disease, cellular movement, etc. (Fig. 3c). Moreover, oxidative phosphorylation, estrogen receptor signaling, hepatic fibrosis signaling, IL-8 signaling were found to be activated in tumors, whereas PD-L1 cancer immunotherapy pathway, the antioxidant action of vitamin C, PTEN signaling and PPAR signaling were inhibited (Fig. 3d). Several factors, e.g., *ATF4, JUNB, PPRC1* and *NFKBIZ*, were found to promote or

inhibit a large number of inflammation-related genes, genes involved in cell proliferation, invasion, cell movement and lipid metabolism (Fig. 3e). Analyses of upstream TFs revealed numerous TFs, e.g., *GATA3, TWIST1, STAT1* and *STAT*2, were significantly activated. In contrast, some TFs, e.g., *SOX7, NFIX,* and *LMO2*, were drastically inhibited (Fig. 3e). Overall, the most affected regulators and effectors appeared to be related to *STAT* genes and participate in the regulation of autophagy, antimicrobial and interferon responses (Fig. 3f; Supplementary Fig. S4a). As the key regulators in metastasis-related EpC clusters (C7, C8) (Supplementary Fig. S4b), STAT1 and STAT2 belong to the STAT family and play essential roles in interferon (IFN)-signaling in response to stimulation by cytokines, growth factors, and hormones[11–13]. Of interest, the expression patterns of *STAT1* and *STAT2* were variable in various cancer types (Supplementary Fig. S4c). In head-neck squamous cell carcinoma (HNSCC), *STAT1,* and *STAT2* expression levels were significantly upregulated in tumors compared to normal tissues and positively correlated with individual cancer stage, grade, and nodal metastasis (Supplementary Fig. S4d). *STAT1* and *STAT2* levels were particularly high in tumors with infiltration of immune cells, including CD4$^+$T, macrophage, neutrophil, and dendritic cells (DCs) (Supplementary Fig. S4e–g). These results suggest that transcriptomic features of malignant cells are indicative of invasion and metastasis of LSCC, and *STAT*1 and *STAT*2 in the TME may participate in tumorigenesis and immune cell infiltration.

**T cell subcluster reprograming landscape in metastatic LSCC.** T cells are the major immune cells that fight cancerous cells and represent the dominant moderator for tumor immunity. As the second most abundant cell type in LSCC, the T cells, may play an important role in tumor immunity. To this end, we further analyzed the T/NK cells by re-clustering them into 10 subclusters, including CD8+ T cells (C0, C2 and C5), CD4+ memory T cells (C1, C4, C6 and C7), CD4+ Tregs cells (C3) and γδ T cells (C8 and C9)[14] (Fig. 4a, b). The high degree of heterogeneity of the T cell populations in LSCC supports the notion that LSCC has a highly complex TME (Fig. 4c). Indeed, the four types of samples displayed distinct cell cluster compositions, whereas the samples without metastasis displayed similar cell cluster compositions (Fig. 4c). The fact that tumor tissues are infiltrated with a greater number of T cells than normal tissues indicates that T cells play an important role in tumorigenesis and development.

To discover the effects of the T cells, we performed GSEA using the top DEGs among all of the T subclusters. GSEA data showed that CD8+ T cells and γδ T cells displayed higher cytotoxicity and antigen-binding activity, but CD8+ T cells showed lower metabolism and proliferation than γδ T cells (Supplementary Fig. S5a, b). Moreover, cluster C5 CD8+ T cells mainly existed in the T samples, cluster C0 and C2 CD8+ T cells and C8 γδ T cells were mostly present in the N samples, and cluster C9 γδ T cells were mostly present in the T samples with metastasis (Fig. 4c). Together, these data suggest that γδ T cells and CD8+ T cells may play a key role of anti-inflammation and anti-tumor, and the CD8+ T cells appear to have been reprogrammed by the LSCC TME, enabling malignant cells to escape immune attacks.

Of interest, a large number of Tregs (C3) were detected in the tumor tissues, further supporting the immunosuppressive state of the LSCC TME (Fig. 4a–c, Supplementary Fig. S5b). Consistently, SCENIC analyses identified several key TFs (e.g., *FOXP3, NFKB complex, SOX4, PRDM1*, etc.), which are known critical regulators of Treg characteristics (Fig. 4d; Supplementary Fig. S5c). The GSEA results identified increased activities of the death receptor, tumor necrosis factor−activated receptor, and regulation of autophagy, suggesting a high level of apoptosis (Fig. 4e; Supplementary Fig. S5d). Therefore, the drastically increased number of Tregs in LSCC reflects a highly immune-suppressive TME. Additionally, high levels of *BATF*, an important regulator of T cells differentiated into Th17 cells[15], were observed in LSCC (Supplementary Fig. S5e). Th17 cells are known to play different roles in different diseases. For example, Th17 cells increase tumor progression by activating angiogenesis and immunosuppressive activities; Th17 cells also drive anti-tumor immune responses by producing IFN-γ[16]. The balance between Th17 cells and Tregs is believed to be critical for regulating cancer autoimmunity[16].

**Aberrant developmental state of B cells in metastatic LSCC.** B cells are responsible for the humoral immunity component of adaptive immunity. Our scRNA-seq detected abundant B cells among the LSCC tissue samples. Re-clustering of the B cells identified 9 subclusters, including memory B cell (cluster C2), naive B cell (clusters C0 and C1), germinal center (GC) B cell (clusters C3) and plasma cell (clusters C4, C8)[14] (Fig. 5a, b). These subclusters showed different distribution patterns among the five types of tissues analyzed (Fig. 5c). The developmental trajectory analyses of B cells revealed that the plasma cells represented the early development stage (Fig. 5d). GSEA detected increased activities in complement activation, protein exit from the endoplasmic reticulum, and SRP−dependent co-translational protein targeting to the membrane in both clusters C4, C8, and upregulated endoplasmic reticulum unfolded protein response in cluster C8, suggesting compromised functions of the plasma cells (Fig. 5e; Supplementary Fig. S6a, b). All the results implied that the B cells infiltrated failed to play the anti-tumor humoral immunity role, which may partially explain the immune escape of LSCC.

To identify key regulators for B cell development in LSCC, we performed GSEA and SCENIC analyses. The pathway enrichment results showed that glutathione metabolism, mRNA surveillance pathway and oxidative phosphorylation were inhibited in plasma cells (Supplementary Fig. S6b). Consistent with their function, this finding indicates that hypoxia and oxidative free radicals (ROS) in the TME might lead to endoplasmic reticulum stress in plasma cells so the function of staphylococcus aureus infection and RIG-I-like receptor signaling pathway remained suppressed (Supplementary Fig. S6b). TF analyses identified that *XBP1* was significantly upregulated in plasma cells, which had been reported to promote the accumulation of unfolded proteins in the endoplasmic reticulum (ER)[17]. TFs regulating cell proliferation, differentiation, and apoptosis were also identified to be differentially expressed, e.g., *KLF10, IRF4, REL, NFKB2*, etc. (Supplementary Fig. S6c). These results suggest that B cells in the LSCC TME display weaker anti-tumor effects due to its endoplasmic reticulum stress and poor development and that TFs, e.g., *STAT1, XBP1,* and *CREB3L2*, may play an important role in regulating B cell vitality in LSCC.

**Myeloid cell subcluster enrichment and reprograming landscape in metastatic LSCC.** Myeloid cells were re-clustered into ten subclusters and the cells were identified as macrophages, neutrophils, monocytes and dendritic cells (DCs) (Fig. 6a). Clusters C0 and C1 represent neutrophils, as these cells showed high levels of *CEACAM1* and *CXCR2*[14] (Fig. 6b). Clusters C2, C4, and C6 cells displayed abundant expression of CD14, CD163, and *APOE*, thus representing macrophages (Fig. 6b). Clusters C3, C5 and C7 are monocytes based on their higher expression levels of *VCAM, FCN1* and *S100A12* (Fig. 6b). Cluster C4 highly expressed *CCR7, FSCN1,* and *LAMP3*, thus representing DCs, the most powerful antigen-presenting cells[14] (Fig. 6b). Also, we found a

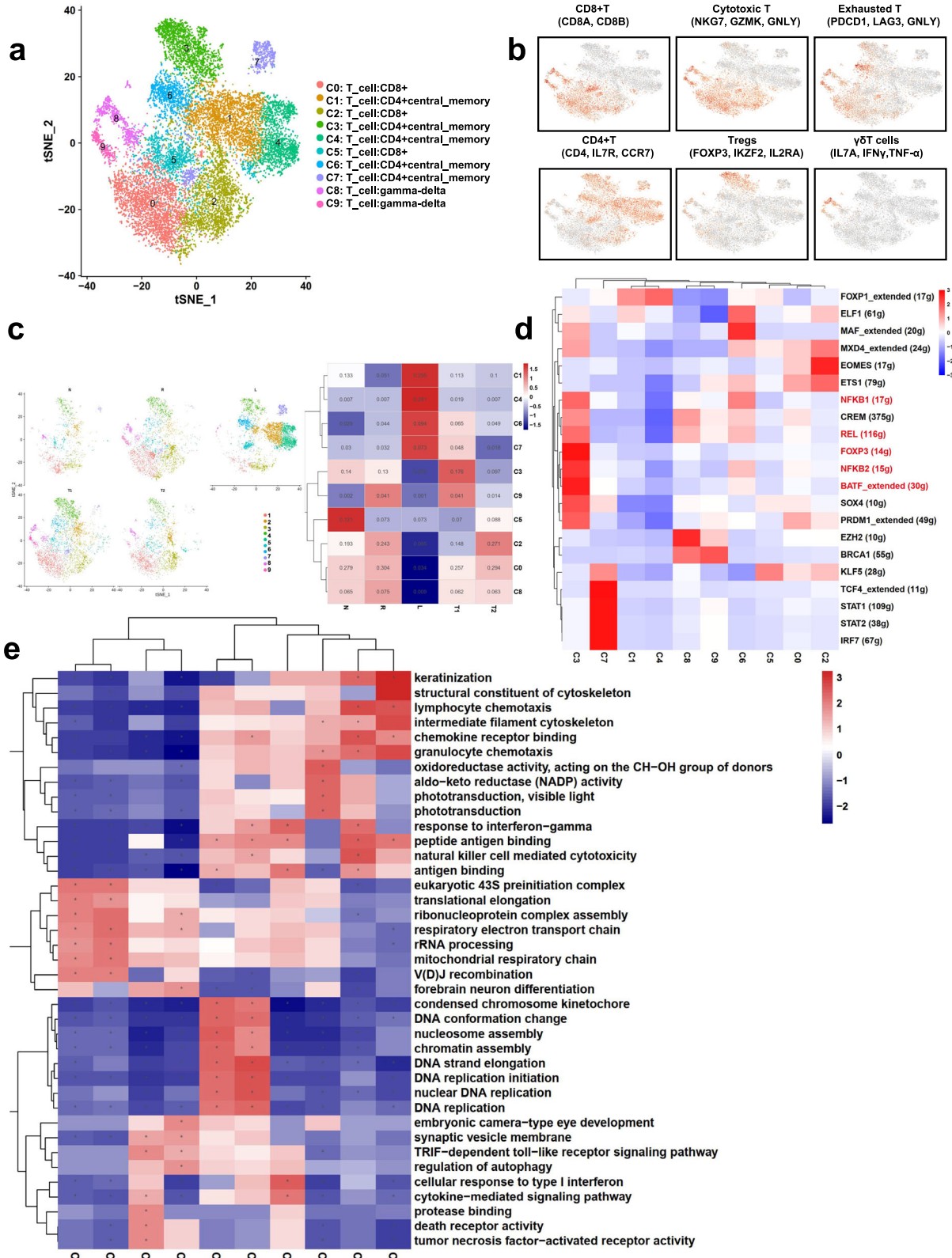

**Fig. 4 T cell heterogeneity in LSCC with lymphatic metastasis. a** t-SNE plots showing the eleven subclusters identified in T cells of LSCC. **b** Marker gene expression in the eleven T cell subclusters. **c** t-SNE plots and heatmap showing the distribution of the eleven T cell subclusters in the four tissue types analyzed in the present study. **d** Heatmap showing the activity of TFs in the eleven T cell subclusters. **e** GSEA results showing pathway activation differences among the 10 T/NK cell subclusters.

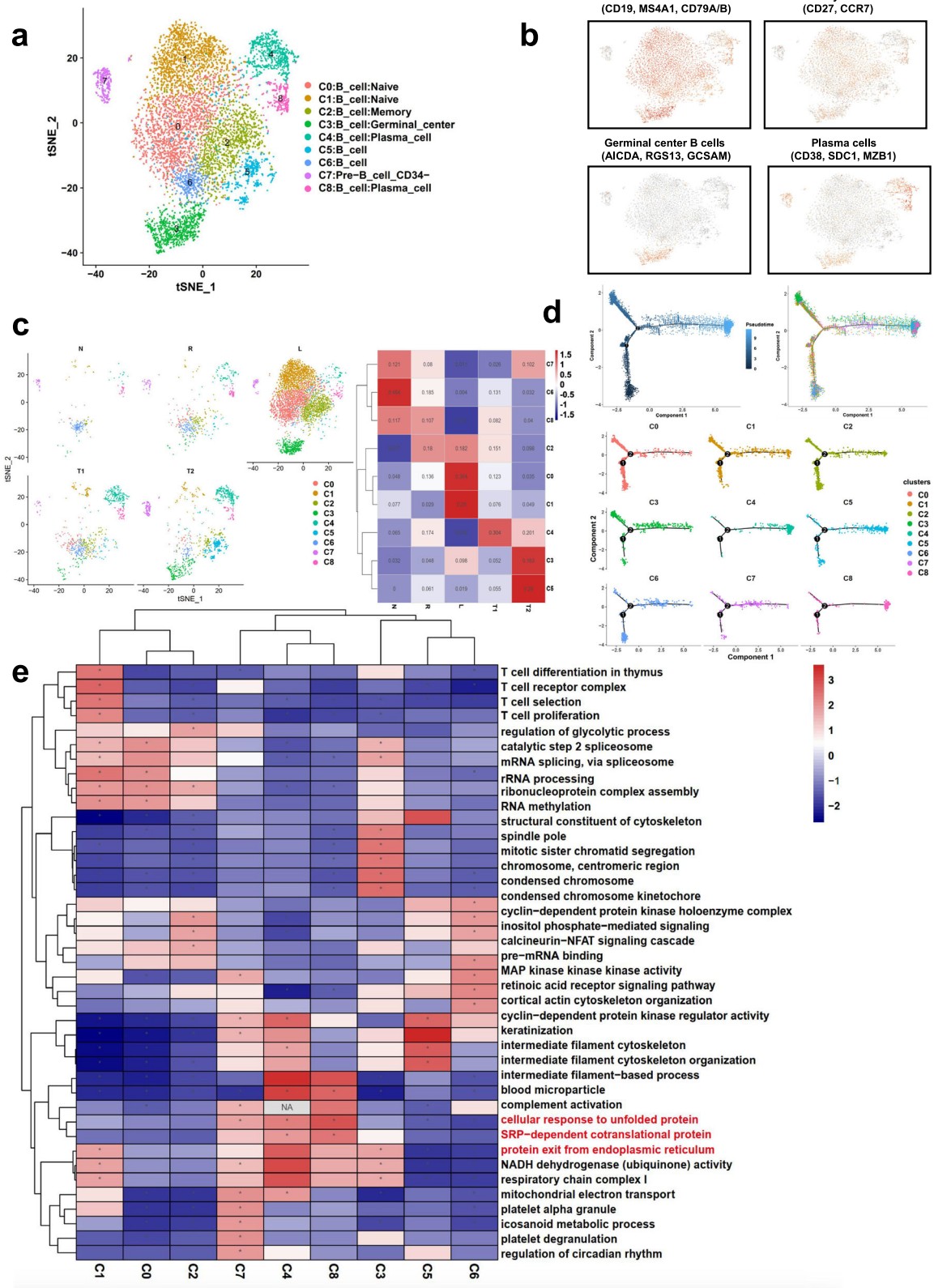

**Fig. 5 B cell heterogeneity in LSCC with lymphatic metastasis. a** t-SNE plot showing the nine subclusters identified in the B cells of LSCC. **b** Marker gene expression in the four major B cell subtypes. **c** Heatmap and t-SNE plots showing the distribution of the nine B cell subclusters in the four tissue types analyzed in the present study. **d** Pseudotime analyses of the developmental trajectory of the B cells detected in LSCC. **e** GSEA results showing pathway activation differences among the nine B cell subclusters.

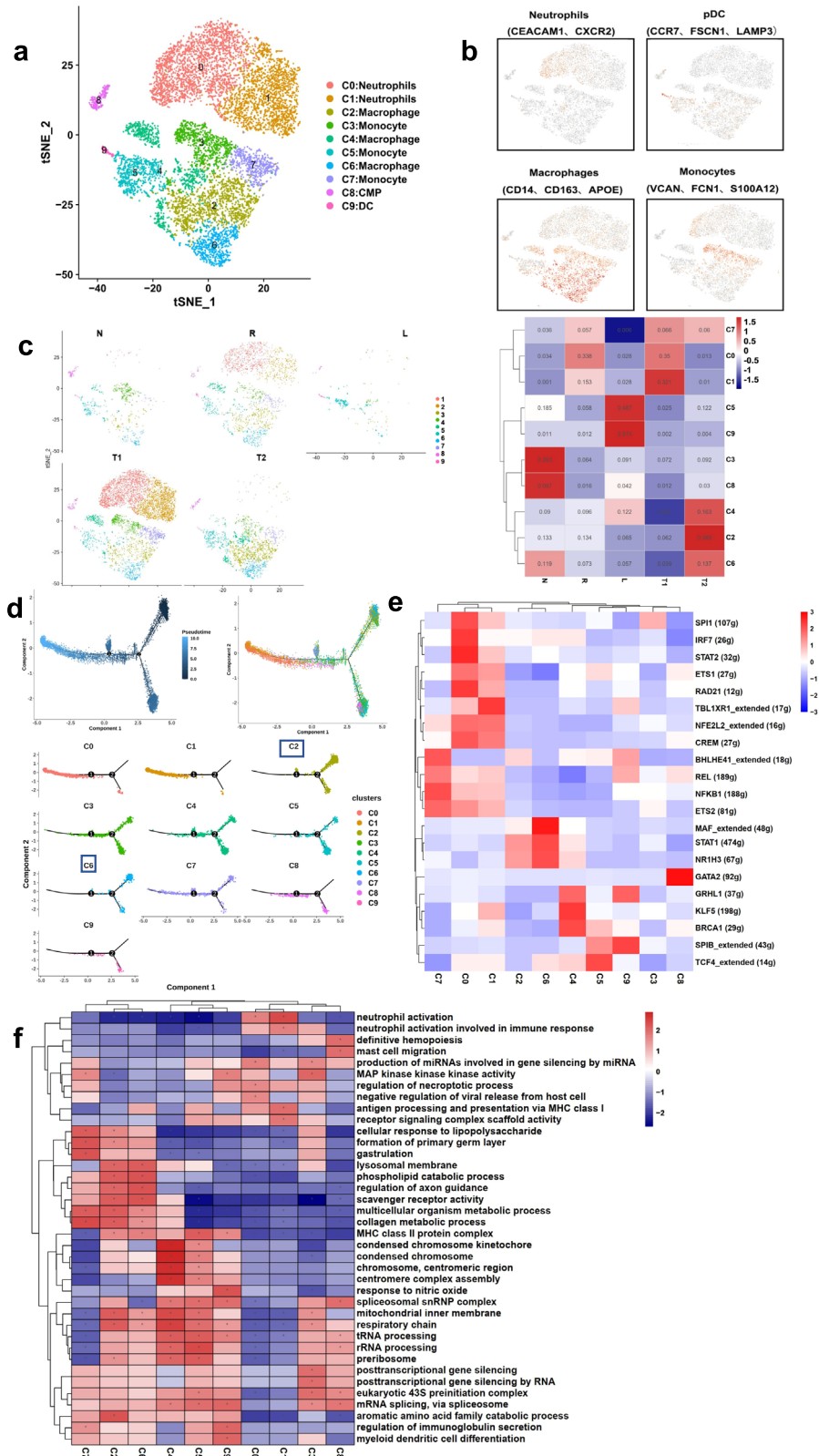

**Fig. 6 Myeloid cell heterogeneity in LSCC with lymphatic metastasis. a** t-SNE plots showing the seven subclusters detected in myeloid cells in LSCC tissues. **b** Marker gene expression in the four major myeloid cell subtypes. **c** Heatmap and t-SNE plots showing the distribution of the ten myeloid cell subclusters in the four tissue types analyzed in the present study. **d** Pseudotime analyses of the developmental trajectory of the myeloid cells detected in LSCC. **e** Heatmap showing the activity of TFs in the seven myeloid cell subclusters. **f** GSEA results showing pathway activation differences among the seven myeloid cell subclusters.

large number of neutrophils enriched in the marginal cancer tissues (R) and tumors with metastasis (T1), reflecting an inflammatory microenvironment along the cancer edges and its importance in tumor infiltration (Fig. 6b, c). In the T2 samples (LSCC in situ without metastasis), macrophages and monocytes, although heterogeneous, are dominant (Fig. 6c).

The developmental trajectory analyses of myeloid cell clusters revealed that the macrophages are at the early developmental stage (Fig. 6d), implying that the infiltrated macrophages fail to play the anti-tumor humoral immunity role, which may partially explain the immune escape of LSCC. The GSEA and SCENIC analyses revealed a number of differentially activated transcription factors and pathways related to interferon-dependent immune responses, differentiation or activation of macrophages, cell cycle, and p53-dependent/independent apoptosis. For example, in neutrophils, *NFKB1*, *CREM*, and *ETS1*, important TFs for cell proliferation, were upregulated significantly (Fig. 6e). These results suggest that neutrophils tend to undergo apoptosis. Meanwhile, levels of *SPI1*, *MAF*, and *STAT1*, which are important TFs involved in the differentiation or activation of macrophages or B cells and interferon-dependent immune responses[18], were increased significantly in C2 and C6 macrophages (Fig. 6e). To reveal the function of myeloid cell clusters, we performed GSEA based on the GO database. The GO terms of "mitochondrial inner membrane respiratory chain, rRNA processing, and tRNA processing" were enriched in macrophages (Fig. 6f), indicating activation (Fig. 6f), suggesting that macrophages in the LSCC TME display significant activation.

**Stromal cell autophagy may promote invasion and metastasis of LSCC**. In the TME of LSCC, stromal cell heterogeneity was less prominent than immune cells, as evidenced by only two cell clusters (one endothelial and one fibroblast) identified (Supplementary Fig. S7a, b). Normal tissue also contained fibroblasts and endothelial cells and fibroblasts were present in marginal LSCC tissue (R) (Supplementary Fig. S7c). GSEA based on DEGs between T and R uncovered enhanced autophagy (Supplementary Fig. S7d, e), suggesting that fibroblasts and endothelial cells in LSCC have an impact on the tumor microenvironment by providing the tumor cells with nutrients, thus promoting invasion and metastasis of LSCC.

**Complex cell-cell communication networks in LSCC**. Interactions among EpCs, fibroblasts, myeloid and endothelial cells were investigated based on the ligand-receptor pairs in the four types analyzed in this study using Cellchat[19]. A complex intercellular communication network appeared to exist among all major cell types in the LSCC TME (Fig. 7a). Many ligand-receptor pairs were detected in the epithelial cells, fibroblast, endothelial cells, and B cells in the TME of LSCC, suggesting potential interactions between any two of these different cell types (Fig. 7b). The tumor cells appeared to interact with most of the cell types in the TME (Fig. 7c). For example, strong interactions existed between the tumor cells and B cells, CD4+ T cells, CD8+ T cells or Tregs through the lymphotoxin and lymphotoxin beta receptor (LTB-LTBR) (Fig. 7c). Moreover, the functions of those ligand-receptor pairs enriched among the four major cell types were analyzed, and the enriched functions included cytokine regulation, immune response and suppression (Fig. 7c, Supplementary Figs. S8–11). The complex cell-cell communication networks in the LSCC TME suggest that the tumor cells have the ability to remodel TME to facilitate tumor immune escape and tumor progression.

## Discussion

Human Cell Atlas (HCA) and Human Tumor Atlas Network (HTAN) projects had been launched, aiming to collect transcriptomic data using scRNA-seq to define key processes and events in the development of human cancers, e.g., the transition from precancerous lesions to malignant tumors[20,21]. scRNA-seq analyses have been used to analyze cell heterogeneity, immune microenvironment, and drug resistance mechanisms of various types of malignancies, including breast cancer, lymphocytes, kidney cancer, renal cell cancers and melanoma[22–25]. One scRNA-seq study on LSCC in situ has been reported[26], but the transcriptomic atlas of LSCC with lymphatic metastasis remains to be determined. In the present study, we analyzed samples from 3 LSCC patients with lymphatic metastasis and 3 LSCC patients without lymphatic metastasis to obtain the single-cell transcriptomic profiles related closely to metastasis via scRNA-seq. Our analyses identified seven major cell types in LSCC, including epithelial-derived cells, T lymphocytes, B lymphocytes, myeloid cells, NK cells, endothelial cells, and cancer-associated fibroblasts, suggesting that immune cells played important roles in the TME of LSCC. The degree of cancer heterogeneity has been correlated with malignant features, including tumor invasion, metastasis, drug resistance, and prognosis[27–29]. The high degree of cellular heterogeneity in LSCC suggests a complex TME of LSCC. Two of the three cell subclusters in LSCC are potentially cancer stem cells (CSCs) because they express higher levels of *SOX2* and *SOX4*. CSCs have been shown to play a critical role in tumor survival, proliferation, metastasis, and recurrence by promoting tumor cell survival through self-renewal and immortal proliferation[30–32]. In addition, the identification of stem cell-like subclusters in lymphatic tissues may account for tumor immune escape in LSCC.

Our data also show that the cell subclusters associated with invasion and metastasis display gene signatures of immune chemotaxis and epithelial-to-mesenchymal transition (EMT), e.g., *STAT1* and *STAT2*. The STAT family members have seven conservative structural features: the N-terminal domain (ND), coiled-coil domain (coiled-coil), DNA-binding domains (DBD), linker domain (Linker), Src homology 2 domain (SH2) and the C-terminal transcriptional activation domains (TAD)[11]. Before being activated, non-phosphorylated STATs bind to each other through the ND domain to form antiparallel dimers, which constantly shuttle between the cytoplasm and nucleus. However, after the receptor is activated by cytokines, the phosphorylation of the STAT proteins by JAK leads to the spatial reorganization of the STAT dimer complex, forming active parallel dimers, which are then separated from the receptor and transferred to the nucleus[33]. STAT1/ STAT2 has a large number of target genes, including *NO*, *BCL-2*, *p21*, and *CCND1*, which all participate in pro-apoptotic and cell-cycle regulation, and act as tumor suppressors in various cancers[33]. Consistently, STAT1 knockout mice display increased susceptibility to experimentally-induced tumors and spontaneously develop mammary adenocarcinomas and ovarian teratomas[34]. Moreover, STAT1 expression and activation are abnormal in malignant pleural mesothelioma, pancreatic cancer, and breast cancer[35–37]. PD-L1 and p-STAT1 have been found co-expressed in breast cancer cells, and high p-STAT1 expression or STAT1 mRNA levels are associated with poor outcomes and advanced clinical stages in breast cancer, suggesting p-STAT1 was related to tumor immune escape[38]. In the present study, both STAT1 and STAT2 are highly expressed and activated in LSCC, and associated with immune cell infiltration, suggesting their pro-cancer function and potential immune microenvironment remodeling functions.

Tumor progression depends on not only the degree of malignant transformation but also the microenvironment in which the tumor is located. A well-accepted hypothesis regards tumor cells as "seeds", and the microenvironment as "soil", and that they interact with each other and evolve together to promote cancer development[39]. Recently, cancer immunotherapy based on the

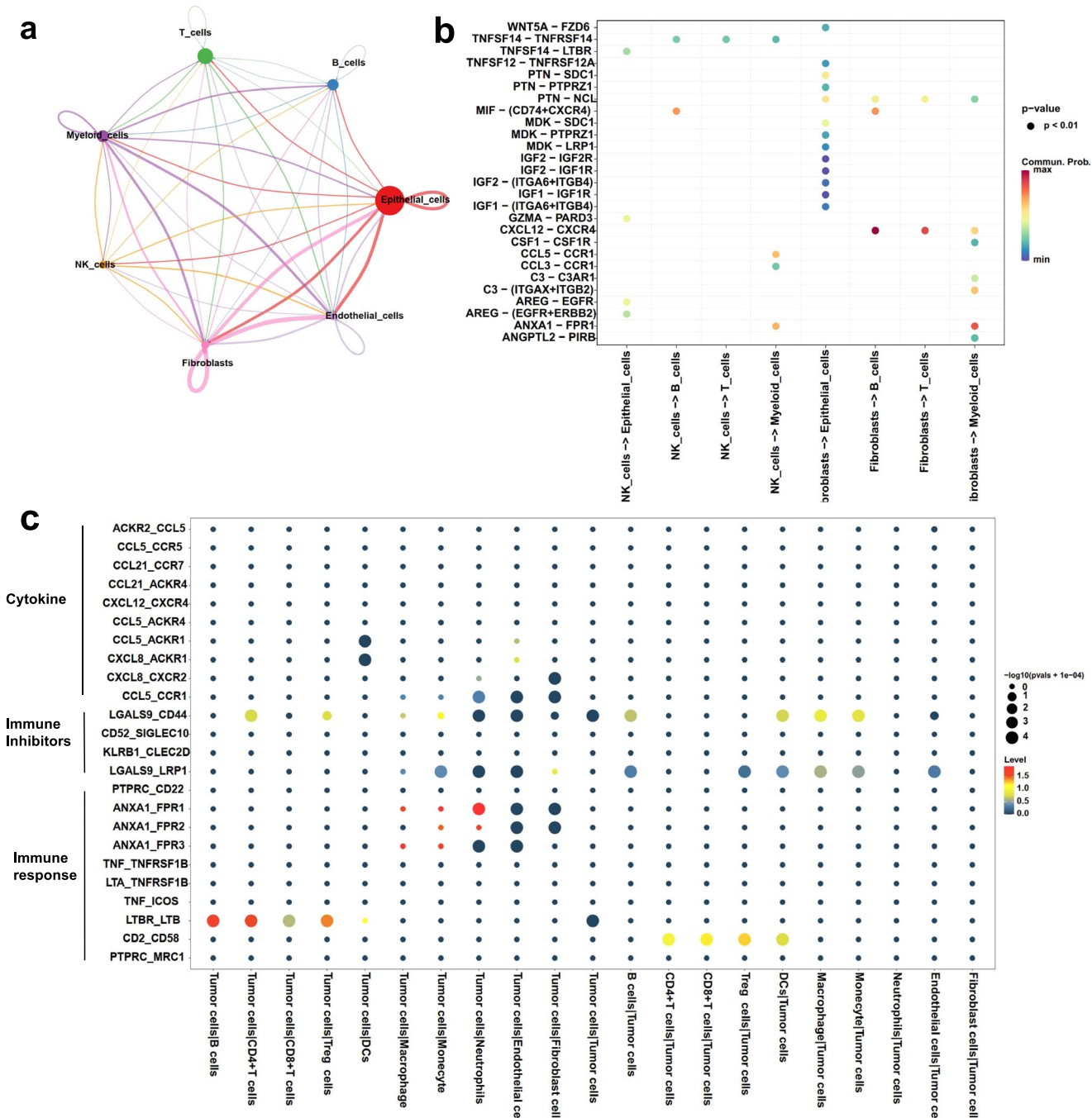

**Fig. 7 Complex intercellular communication networks in the LSCC TME. a** Schematic illustration of the cell-cell interaction networks among all major cell types identified in the LSCC TME. **b** Plots showing the dominant ligand-receptor pairs expressed between eight cell-cell pairs in the LSCC TME. **c** Plots showing selected interactions between the tumor cells and other major immune cell types in the LSCC TME. The interactions are mediated through ligand-receptor pairs known to have immune functions.

immune microenvironment has developed rapidly, especially the immune checkpoint inhibitor therapy[40,41], e.g., CTLA-4, PD-1, etc. Although this therapeutic method has been proven promising, the effects seem very uneven among different patients or different cancer types. Factors such as the cell-type composition, developmental stage, and metabolic states in the tumor immune microenvironment all affect the effectiveness of immunotherapy[42]. The presence of a large number of T cells, especially Tregs, abnormal plasma cells, neutrophils and M2 polarized macrophage cells in LSCC, indicates an immunosuppressive and pro-inflammatory microenvironment.

Tregs function to regulate or suppress other cells in the immune system by controlling the immune response to self and foreign antigens and helping prevent autoimmune diseases. In malignant tissues, the immunosuppressive effect of Tregs is one of the key factors for tumor immune escape. Treg-mediated immune tolerance is also closely related to tumor metastasis and may serve as a potential target for immunotherapy[43]. In LSCC, the homeostasis of T cells is inhibited, whereas the apoptosis and autophagy pathways are activated, suggesting that Tregs tend to undergo apoptosis and autophagy. Traditionally the decrease of Tregs would lead to the relief of anti-tumor immunosuppression.

However, it has been reported that the apoptotic Tregs are more suppressive because apoptotic Tregs eliminate PD-L1 blockade-mediated anti-tumor T cell immunity[44]. In addition, the apoptotic Tregs could quickly convert ATP into adenosine targeting T cells and bind the receptors on the surface of T cells to affect the T cell functions. Several reports have proposed that the role of Tregs in cancers might be associated with inflammation[45,46]. Given that Tregs appear to cause immunosuppression and promote tumor progression in LSCC, immune therapy might work on LSCC as well.

B cells are derived from bone marrow and can differentiate into plasma cells after antigen stimulation, followed by synthesis and secretion of antibodies to achieve humoral immunity. In cancerous tissues, B cells can infiltrate and form tertiary lymphoid structures (TLSs) or germinal centers (GC), but the effect of tumor-infiltrating B cells has not been fully understood. A study on lung cancer found the adoptive transfer of B cells to tumor-draining lymph nodes can cause tumor regression in a mouse model of breast cancer with lung metastasis[47]. Another study also found that B cells in the tumor-draining lymph node can be recruited to tumors, lungs, and secondary lymphoid organs in vivo and can directly kill tumor cells through the Fas/FasL pathway[48]. Analyses of the composition of immune cells in tumor tissues from metastatic melanoma and renal cell carcinoma cohorts have found that B cells and TLSs are positively correlated with patients' responses to immunotherapy[49]. In LSCC, more B cells are present in cancerous tissues than in normal tissues. However, effector B cells (i.e., plasma cells) infiltrated LSCC tissues are at the early stage of development and display enhanced endoplasmic reticulum unfolded protein response and ubiquitin-dependent protein catabolic process, which may explain the compromised humoral immunity. Although the exact function of B cells in LSCC metastasis remains unknown, these B cells may represent a good target for developing laryngeal cancer immunotherapy.

Derived from common myeloid progenitor cells, myeloid lineage cells include neutrophils, basophils, eosinophils, erythrocytes, macrophages, monocytes, dendritic cells, granulocytes, and megakaryocytes (platelets), which are the main component of the natural immune system and serve as the first line of defense against infection[50]. In addition to lymphocytes, myeloid lineage cells are also one of the important components of tumor-infiltrating immune cells, and play an important role in regulating tumor inflammation, angiogenesis, and immune cell activation[51,52]. Among monocytes, neutrophils, macrophages and DCs, macrophages and monocytes are enriched in tumor tissues, whereas neutrophils are dominant in marginal cancer tissues, and most of the infiltrating macrophages in LSCC are abnormal developing states. In addition to macrophages, a large number of infiltrated neutrophils are present in tumor tissues, consistent with their roles in the activation of tumor cell inflammation and chemokine-related pathways. Inflammation in TME is a "double-edged sword". On the one hand, inflammatory factors, in general, can kill pathogens, promote tissue repair, and prevent tumor growth. On the other hand, inflammation can promote tumor growth and progression by promoting angiogenesis and metastasis, subverting the anti-tumor immune response, and changing the sensitivity of tumor cells to chemotherapeutic drugs[53]. Although the neutrophil-mediated antibacterial humoral response is activated in LSCC TME, proper regulation of T and B cell differentiation seems inhibited. Consequently, the role of neutrophils in LSCC TME tends to promote tumor progression.

Stromal cells in the TME of LSCC include fibroblasts and endothelial cells. While their number is smaller than immune cells, they appear to have more interactions with tumor cells. Stromal cells in the tumor microenvironment regulate tumor growth, metastasis and proliferation, and this has been verified by numerous studies on various cancers, including hepatocellular carcinoma, pancreatic carcinoma, head and neck carcinoma, colorectal carcinoma and lung carcinoma[54–59]. Indeed, compared to the center of LSCC, the stromal cells in marginal tissues display enhanced autophagy, suggesting a potential role in promoting tumor invasion and infiltration.

In summary, the present study not only provided a comprehensive atlas of LSCC but also revealed the complex TME of LSCC. Data reported here will inspire future studies on the molecular mechanism underlying LSCC invasion and metastasis and also facilitate the efforts in developing early diagnostics and effective therapeutics for LSCC.

## Methods

**Sample collection.** Laryngeal squamous cell carcinoma (LSCC) in situ (T), normal laryngeal mucosal epithelial tissue adjacent to LSCC (N), marginal tissue of LSCC (R), lymph nodes with metastatic LSCC (L) from 3 LSCC patients with lymphatic metastasis and LSCC in situ (T) from 3 LSCC patients without lymphatic metastasis were collected from the Department of Otolaryngology, the Fourth People's Hospital of Shenyang City. All the patients gave their informed consent, and the study was approved by the Institutional Review Board of China Medical University in accordance with the Declaration of Helsinki. All ethical regulations relevant to human research participants were followed. The pathological type of each tissue was confirmed by at least two well-trained pathologists, and the clinical characteristics are shown in Supplementary Table S1. Immediately after surgical removal, the tissue samples were dissected into two segments: one was digested into a single-cell suspension for scRNA-seq; the other one was immediately transferred into 4% paraformaldehyde for fixation and paraffin embedding for immunohistochemistry.

**Preparation of single-cell suspensions.** A tissue sample submerged in a digestion solution containing Type II and Type IV collagenase (Type II and Type IV from Gibco, Cat#17101015, 17104019) was cut into smaller pieces (3 mm × 3 mm) which were then transferred to a 50 ml centrifuge tube containing 5 ml of the digestion solution followed by incubation in a 37 °C water bath for 30 min. After digestion, the digestion solution was passed through a pre-wet cell sieve (70 μM, BD, cat# 431751), and the filtrate was collected into a new 50 ml centrifuge tube for centrifugation at $900 \times g$. An aliquot of 3 ml precooled erythrocyte lysate was added to the cell pellet to resuspend the cells followed by incubation at room temperature for 3 min. After centrifugation ($300 \times g$ at room temperature for 10 min), an appropriate volume of DPBS containing 2% FBS (20 μl/ml) was added to resuspend the cells to make the final concentration at >10,000 cells per ml. To qualify for downstream scRNA-seq analyses, a single-cell suspension was confirmed to contain neither small cell clusters with more than two cells adhered to each other, nor cell debris and other particulate matters, and the cell viability was >80%.

**scRNA-seq library preparation and sequencing.** The 10X Genomics Chromium Single Cell Platform was used for scRNA-seq. scRNA-seq libraries were generated using the Chromium Single Cell 30 Library and Gel Bead Kit v2 (10X Genomics, PN-120237) and the Chromium TM Single Cell A Chip Kit (10X Genomics, PN-120236) following the manufacturer's protocols. The cDNA concentration was measured using a Qubit 4 Fluorometer (Thermo Fisher, Cat#Q33238) and the fragment sizes were determined by an Agilent 2100 Bioanalyzer (Agilent, Santa Clara, CA, USA). All samples were sequenced at multiplex

paired-end 150 bp on an Illumina NovaSeq 6000 sequencer with 100 G high quality of sequence reads.

**scRNA-seq data processing**. The sequencing data were converted to cell expression matrix using cell Ranger software, and quality control and statistics of the raw data were carried out for exon ratio, second-generation sequencing Q30, barcode and UMI numbers. Single-cell sequencing results of different samples were combined, and the batch differences due to experiments or sequencing and the differences in sequencing depth, UMI expression, gene expression and the proportion of mitochondrial ribosomes were removed. The following cells were also filtered out: (1) those with a total gene number of >6000 or <200; (2) those with a mitochondrial gene ratio of >20%, and (3) those with a hemoglobin gene ratio of >1%.

To evaluate cell-cycle effects, the cell-cycle score of each cell was calculated based on the expression of cell-cycle genes followed by principal clustering analysis (PCA). If the cell-cycle effect was too large, it was then removed by linear regression. For PCA dimensionality reduction and clustering, the "scaledata" function in the Seurat package was used to normalize the data of hypervariable genes. Louvain algorithm was used to cluster the cells through t-SNE function in the Seurat package.

**Annotation of major cell types and their subclusters**. Wilcoxon algorithm was used to analyze the marker genes for all clusters by scoring the marker genes with one vs rest algorithm, as described previously[14]. The genes with highly specific expression in each cluster, logFC >0.25 and expressed in at least 20% of the cells were selected as the significant marker gene of the cluster. The cell type was annotated using single R on the basis of marker genes, with the normal immune cell expression profile as the reference database. The expression status of specific genes or gene sets in each cell was analyzed through LOUPE software to further identify cell subsets. Specifically, *CD8A, NKG7, GZMB, GZMH* and *GNLY* labeled CD8+ T cells; *CD4, IL7R* and *CCR7* labeled CD4+ T cells; *FOXP3, KLRB1, IKZF2* and *TNFRSF4* labeled Tregs; *KLRF1, XCL1* and *XCL2* labeled NK cells; *CD19, MS4A1, CD79a, and CD49b* labeled B cells; *CD27* and *CCR7* labeled memory B cells; *AICDA, RGS13*, and *GCSAM* labeled GC B cells; *CD38, SDC1*, and *MZB1* labeled plasma cells; *CEACAM1* and *CXCR2* labeled neutrophils; *CCR7, FSCN1*, and *LAMP3* labeled DC cells, *CD14, CD163*, and *APOE* labeled macrophages; *VCAN, FCN1*, and *S100A12* labeled monocytes. The cell differentiation trajectories pseudotime analyses were performed using Monocle in R package (https://cole-trapnell-lab.github.io/monocle3).

**Gene set enrichment analysis (GSEA)**. Human GO and KEGG datasets as the target gene sets were downloaded from the GESA websites to prepare the three files that the GSEA software[9] needed (.gct file - Gene expression scale; .cls file - information table of each subgroup; .rnk file - gene sequencing list). The enrichment fraction of each subgroup in different functions/pathways in GSEA was imported, and the functional and activated signaling pathways of the subgroup were analyzed. As the output images of GSEA software were of poor quality and not intuitive enough, we imported the results into R Studio to replot the heatmaps.

**Single-cell regulatory network inference and clustering (SCENIC) analysis**. SCENIC, a computational method for simultaneous gene regulatory network reconstruction and cell-state identification from scRNA-seq data[10], was adopted in the present study. The data from SCENIC analyses were imported into the R Studio (pheatmap) to generate the heatmaps.

**Cell-cell communication analysis**. CellChat[19] was used for analyzing intercellular communication networks based on the receptor-ligand interactions from multiple databases. The scRNA-seq input data for CellChat included quantitative count data and cell-type annotation information. In brief, the percentage and the average of gene expressing for each gene in the cells were calculated. The ligand-receptor pairs were filtered to obtain receptor and ligand genes exceeding a specified threshold (the default is 10%). A pair-to-pair comparison was then performed between all cell types in the dataset, and the actual mean value of the ligand-receptor pairs between two cell types was calculated to speculate *p*-value of the receptor-ligand pair in 2 cell types. Finally, the highly specific interactions between cell types were arranged through the enrichment results of significant ligand-receptor pairs.

**Immunofluorescence**. Paraffin sections (4 μm in thickness) of the four types of tissues (T, L, R, and N) were used for immuno-fluorescent staining. Antigen retrieval was performed by boiling the slides with sections in Citrate Antigen Retrieval Solution (pH 6.0) for 5 min in a microwave, followed by blocking with the goat serum for 15 min at room temperature. The first antibodies used included EPCAM (for epithelial-derived cells at a dilution of 100X, Abcam, Cat#ab223582), CD3 (T cells at a dilution of 50×, Abcam, Cat#ab135372), CD19 (for B cells at a dilution of 50×, Abcam, Cat#ab245235), CD33 (myeloid cells at a dilution of 50×, Abcam, Cat#ab269456), CD56 (NK cells at a dilution of 50×, Abcam, Cat#ab220360), VEGFC (for endothelial cells at a dilution of 50×, Abcam, Cat#ab83905), a-SMA (cancer-associated fibroblasts at a dilution of 200×, Cell Signaling Technology, Cat#19245), STAT1 (diluted at 100×, Abcam, Cat#ab239360), STAT2 (diluted at 100×, Abcam, Cat#ab32367), OCT4 (stem cells at a dilution of 100×, Abcam, Cat#ab181557), SOX2 (stem cells at a dilution of 100×, Abcam, Cat#ab97959). CoraLite488 or Cor-aLite594 Conjugated Antibodies were used as secondary antibodies, and DAPI was used to counter-stain the nucleus. Immunofluorescent images were taken using a microscope (Nikon, A1R) equipped with imaging software (NIS-Elements Viewer, 5.21.00_b1483, Nikon).

**Copy number variation (CNV) and benign/malignant analysis of epithelial-derived cells**. The CNV analyses were carried out via importing the data, including the original matrix of scRNA-seq, the reference, and gene or chromosome location information into the inferred CNV software to analyze the CNV information by comparing the gene expression value in each of cell types with the reference cell (i.e., normal immune cells). The benign/malignant analysis of each cell was also based on the CNV value compared to the reference value by scCancer software, and the cells are scored based on the changes in overall gene copy number (i.e., malign score).

**Ingenuity pathway analysis**. From the differentially expressed genes in the epithelial-derived cells between groups T and N, 3,000 genes with the largest difference were selected and uploaded to IPA (Qiagen, https://www.qiagenbioinformatics.com/products/ingenuity-pathway-analysis/). The canonical pathway, interaction network, disease, functions, and upstream regulatory factors were analyzed.

**Statistics and reproducibility**. R and R Studio software (version 4.0.2, R Foundation for Statistical Computing, Vienna, Austria; http://www.r-project.org) ware used for statistical analysis and data visualization including Louvain algorithm, SCENIC, CNV analyses, CellChat. Z-score and *P*-value were used to judge statistically significant. |z-score|>2 is considered meaningful. The

positive and negative Z-scores indicate activated and suppressed molecular interactions, respectively. *P*-values < 0.05 were considered statistically significant.

**Reporting summary**. Further information on research design is available in the Nature Portfolio Reporting Summary linked to this article.

## Data availability

The scRNA-seq data have been deposited into the National Center for Biotechnology Information Sequence Read Achieve database (accession no. GES206332). All source data behind graphs in Figs. 1–7 can be found in Supplementary Data 1. Any other data are available from the corresponding author upon reasonable request.

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

## Acknowledgements

This work was supported by the National Natural Science Foundation of China (81372876 to WF), Liaoning Science and Technology Project (2021JH6/10500157 and LQNK201726 to WF and YS). Shanghai Biotechnology Corporation is acknowledged for its help with sequencing and bioinformatic analyses of the data.

## Author contributions

W.Y. and W.F. designed the research. Y.S. performed the experiments. Y.S., W.F., and W.Y. analyzed the data. S.C., Y.L., and Z.X. helped collect samples and provided the clinical data. Y.S. and W.Y. wrote the manuscript. All authors have read and approved the final submitted manuscript.

## Competing interests

The authors declare no competing interests.

## Ethical approval

All patients gave their informed consent, and the study was approved by the Institutional Review Board of China Medical University in accordance with the Declaration of Helsinki.
