## [Peer review file · Communications Biology]

Reviewers' comments:

Reviewer #1 (Remarks to the Author):

Paper # COMMSBIO-22-4201-T entitled "The Tumor Microenvironment of Laryngeal Squamous Cell Carcinoma Revealed by Single-Cell Transcriptomic Analyses" carried out bioinformatics analysis on a new scRNA-seq atlas of LSCC.

This is a general study for LSCC by scRNA-seq and bioinformatics, and is relevant to cancer and metastasis.

Several issues should be paid careful attention and addressed before publication consideration:

1. The organization of samples is not solid enough. There are only 3 patients with lymphatic metastasis included. Actually, the patients without lymphatic metastasis should also be included as a key and necessary compared group.
2. The evidences of used cell type markers should be cited fully.
3. Authors used immunofluorescent staining to validate a few marker proteins of known cell types. It is ok. But, the important and expected work should be using immunofluorescent staining to validate the identified new (tumor) cell types and their new markers.
4. In addition to typical GSEA used in bulk sample analysis, ssGSEA should be used to reveal more functions related to individual cells.
5. CellPhone was used to analyze cell-cell communication. It is suggested to apply some new approach like cellchat to obtain results.
6. Authors stated that, this work will help early diagnostics and effective therapeutics for LSCC, however, no significant results were supplied for such key conclusion. The detail candidate markers should be reported, and validated by immunofluorescent staining, and evaluated in public data with large samples.
7. It is suggested to add a new final figure to summarize the findings and validations in this work, and highlight their clinical values.

Reviewer #2 (Remarks to the Author):

The authors performed single-cell analyses of primary tumors, lymph node metastases, and surrounding non-malignant tissue from three patients with laryngeal cancer to characterize the frequency of cell populations, the signaling pathways, and the microenvironment characterizations potentially involved in laryngeal cancer metastasis. The study design is well structured and takes advantage of single-cell analysis to deeply characterize cell subpopulations, signaling pathways and cell-to-cell communication. On the other hand, the weakness of this study is that a single-cell analysis in a small number of cases was not able to truly verify the invasive metastasis mechanism, and some of conclusions in the result sections have overclaims. My comments are as follows.

1. The title should be informative. What was revealed by scRNAseq?
2. In line 180, the subtitle, "Regulatory T cells (Tregs) play an important role in LSCC invasion and metastasis" is not proved by data in this study. Differential Treg/Th17 ratios between primary and metastatic sites do not mean that Tregs contribute to metastasis and particularly, invasion. Ultimately, the roles in invasion and metastasis should be proven by in ex vivo or in vivo experiments. The authors should include a reasonable subtitle and conclusion, simply describing the predominance of immunosuppressive fraction of immune cells in metastases.
3. In analogous to the comment #2, "promotion of invasion and metastasis" in line 246 and line 283 is not fully supported by data. The authors should include a reasonable subtitle and conclusion.
4. The text in the figures is too small in terms of readability, and should be enlarged as much as possible. The layout could be modified to make the font size larger. Shared legends can be placed

in one place.

Dear Reviewers,

Thanks for giving us an opportunity to revise our manuscript! Following the reviewers' comments, we performed additional experiments and revised the text. Please find our point-by-point responses detailed below. We hope that our revised manuscript is now ready for acceptance.

Reviewer #1:

Question 1: The organization of samples is not solid enough. There are only 3 patients with lymphatic metastasis included. Actually, the patients without lymphatic metastasis should also be included as a key and necessary compared group.

Answer: It is a good suggestion. We added 3 samples of laryngeal cancer without metastasis and performed scRNA-seq, and our new data have been added to the revised manuscript (page 4, lines 101-126).

Question 2. The evidence of used cell type markers should be cited fully.

Answer: The cell type markers were downloaded from Cellmarkers database. Following your advice, we listed all the cell markers used in clustering and added references (Page 5, lines 116; Page 7, lines 192; Page 9, lines 228; Page 10, lines 258).

Question 3. Authors used immunofluorescent staining to validate a few marker proteins of known cell types. It is ok. But, the important and expected work should be using immunofluorescent staining to validate the identified new (tumor) cell types and their new markers.

Answer: Thank you for your suggestion. We performed immune staining using two stem cell markers (SOX2 and OCT4) to validate the CSCs highly relevant to cancer metastasis (Supplementary Figure S3).

Question 4. In addition to typical GSEA used in bulk sample analysis, ssGSEA should be used to reveal more functions related to individual cells.

Answer: It is a good suggestion, but we performed GSEA by using special feature genes in each cell cluster to annotate the main functions of each cell cluster. The ssGSEA is intended for showing the different functions of cell clusters from each sample, which does not serve our

purposes.

Question 5. CellPhone was used to analyze cell-cell communication. It is suggested to apply some new approach like cellchat to obtain results.

Answer: Agreed. We used cellchat to obtain new communication networks, as showed in Figure 7.

Question 6. Authors stated that, this work will help early diagnostics and effective therapeutics for LSCC, however, no significant results were supplied for such key conclusion. The detail candidate markers should be reported, and validated by immunofluorescent staining, and evaluated in public data with large samples.

Answer: Agreed. We evaluated candidate markers STAT1/2 by examining their expression levels, relationship with copy number variations (CNV), immune cell infiltration, cancer grades, stages, and nodal metastasis in HNSCC with public data with larger sample size, and immunofluorescent staining results, as showed in Supplementary Figure S4.

Question 7. It is suggested to add a new final figure to summarize the findings and validations in this work and highlight their clinical values.

Answer: Agreed. We modified the summary hypothesis diagram as suggested.

Reviewer #2

Question 1. The title should be informative. What was revealed by scRNAseq?

Answer: Thank you for your suggestion. We changed the title to “Single-Cell Transcriptomic Analyses of Tumor Microenvironment and Molecular Reprograming Landscape of Metastatic Laryngeal Squamous Cell Carcinoma.”

Question 2. In line 180, the subtitle, “Regulatory T cells (Tregs) play an important role in LSCC invasion and metastasis” is not proved by data in this study. Differential Treg/Th17 ratios between primary and metastatic sites do not mean that Tregs contribute to metastasis and particularly,

invasion. Ultimately, the roles in invasion and metastasis should be proven by in ex vivo or in vivo experiments. The authors should include a reasonable subtitle and conclusion, simply describing the predominance of immunosuppressive fraction of immune cells in metastases.

Answer: Agreed. We revised the text by adding a new subtitle (“**T cells subclusters reprogramming landscape of metastatic laryngeal squamous cell carcinoma**”) reflecting the predominance of immunosuppressive fractions of immune cells in metastases (Page 7, lines 186).

Question 3. In analogous to comment #2, “promotion of invasion and metastasis” in line 246 and line 283 is not fully supported by data. The authors should include a reasonable subtitle and conclusion.

Answer: We added a new subtitle and conclusion(**Myeloid cells subclusters enrichment and reprogramming landscape of metastatic LSCC**”), as suggested (Page 10, lines 253).

Question 4. The text in the figures is too small in terms of readability, and should be enlarged as much as possible. The layout could be modified to make the font size larger. Shared legends can be placed in one place.

Answer: We modified the figures as suggested.

Thank you!

Wei Yan M.D., Ph.D.

Senior Investigator

The Lundquist Institute at Harbor-UCLA Medical Center

Professor of Medicine

David Geffen School of Medicine at UCLA

REVIEWERS' COMMENTS:

Reviewer #1 (Remarks to the Author):

Authors have responded to all my concerned questions, and made a revision well.

One point is that, the data GES206332 cannot be found in NCBI GEO. Authors should address this little issue of their data share before publication.

Reviewer #3 (Remarks to the Author):

My comments have been sufficiently addressed by the authors.

Responses to reviewers' comments:

Reviewer #1 (Remarks to the Author): One point is that the data GES206332 cannot be found in NCBI GEO. Authors should address this little issue of their data share before publication.

Reply: We inadvertently set the release date of dataset GES206332 at 3 years post-deposition when we uploaded data to the GEO database. We have submitted a request for immediate release of this dataset.